# OpenReview forum: "Red Teaming Multimodal Language Models: Evaluating Harm    Across Prompt Modalities and Models"
_EurIPS.cc/2025/Workshop/UPLB — UPLB2025_

### Official Review · Reviewer_Puos · 2025-10-25
**A benchmark to assess the robustness of multimodal LLMs against adversarial prompts for illegal, misleading, and unethical content generation**

**Rating:** 6
**Confidence:** 3

**Review:**

This work introduces a human-curated dataset of adversarial prompts specifically designed to elicit harmful responses—including illegal activities, disinformation, or unethical behavior—from multimodal large language models (MLLMs). The dataset comprises both text-only and multimodal prompts. We evaluate the responses of four widely used MLLMs, which are human-annotated along a scale ranging from not at all harmful to extremely harmful. The statistical analysis reveals a substantial variation in harmlessness across models, indicating that some MLLMs fail to resist adversarial attacks effectively. Contrary to initial expectations, the results also show that text-only adversarial prompts tend to be more effective at inducing harmful outputs than multimodal ones.

The paper is technically sound and presents interesting conclusions. The proposed benchmark could be a valuable contribution to research on fairness in machine learning, which aligns with one of the workshop’s themes. However, the work focuses exclusively on evaluating the outputs of current multimodal large language models rather than developing new theoretical insights or methodological advances to improve fairness or mitigate harmful responses. As such, its relevance to the workshop’s main topic appears marginal, and it is unclear whether it would stimulate substantial discussion among participants.

---

### Decision · Program_Chairs · 2025-11-03

Accept (Poster)